# Ancestral acetylcholine receptor β-subunit forms homopentamers that prime before opening spontaneously

Christian JG Tessier[1,2], Raymond M Sturgeon[1,2], Johnathon R Emlaw[1,2], Gregory D McCluskey[1,2], F Javier Pérez-Areales[1,2†], Corrie JB daCosta[1,2]*

[1]Department of Chemistry and Biomolecular Sciences, University of Ottawa, Ottawa, Canada; [2]Centre for Chemical and Synthetic Biology, University of Ottawa, Ottawa, Canada

**Abstract** Human adult muscle-type acetylcholine receptors are heteropentameric ion channels formed from two α-subunits, and one each of the β-, δ-, and ε-subunits. To form functional channels, the subunits must assemble with one another in a precise stoichiometry and arrangement. Despite being different, the four subunits share a common ancestor that is presumed to have formed homopentamers. The extent to which the properties of the modern-day receptor result from its subunit complexity is unknown. Here, we discover that a reconstructed ancestral muscle-type β-subunit can form homopentameric ion channels. These homopentamers open spontaneously and display single-channel hallmarks of muscle-type acetylcholine receptor activity. Our findings attest to the homopentameric origin of the muscle-type acetylcholine receptor, and demonstrate that signature features of its function are both independent of agonist and do not necessitate the complex heteropentameric architecture of the modern-day protein.

*For correspondence:
cdacosta@uottawa.ca

Present address: †Yusuf Hamied Department of Chemistry, University of Cambridge, Cambridge, United Kingdom

Competing interest: The authors declare that no competing interests exist.

## Editor's evaluation

This compelling study outlines how phylogenetic reconstruction can yield important insights into evolutionary aspects of assembly and function in acetylcholine receptors (AChRs). The authors elegantly demonstrate that ancestral AChR β subunits can form homomeric channels that share important functional hallmarks with their modern cousins, including unliganded gating. The work provides an intriguing framework to evaluate the evolution of the broader family of pentameric ligand-gated ion channels.

## Introduction

Ligand-gated ion channels convert chemical signals into electrical impulses by coupling the binding of small molecules to the opening of an ion-conducting transmembrane pore (*Hille, 2001*). Of the many types of ligand-gated channels, the superfamily of pentameric ligand-gated ion channels (pLGICs) is the largest and most structurally and functionally diverse (*Jaiteh et al., 2016*). Formed from five identical or homologous subunits arranged around a central ion-conducting pore, pLGICs are found in almost all forms of life (*Corringer et al., 2012*). In prokaryotes, pLGICs appear to be exclusively homopentameric, while in eukaryotes both homo- and heteropentameric channels are common. Humans express more than 40 different pLGIC subunits, which are subdivided based on whether they form cation-selective channels activated by acetylcholine or serotonin (5-hydroxytryptamine [5-HT3]), or anion-selective channels activated by γ-aminobutyric acid or glycine. This repertoire of pLGIC subunits, combined with their

**Figure 1.** Subunit composition of heterologously expressed acetylcholine receptors (AChR). Subunit stoichiometry and arrangement of the human adult muscle-type AChR (left), where the agonist-binding sites at the α–δ and α–ε subunit interfaces are indicated with asterisks (*). A reconstructed ancestral β-subunit (β$_{Anc}$; purple) forms hybrid AChRs (middle) where β$_{Anc}$ substitutes for the human β-subunit (β; orange) and supplants the human δ-subunit (δ; green). The principal (+) and complementary (−) interfaces of β$_{Anc}$ must be compatible for two β$_{Anc}$ subunits to sit side-by-side (red highlight), which predicts that homomers formed from multiple β$_{Anc}$ subunits should be possible (right, boxed).

ability to form homo- and heteropentamers, leads to a wealth of structural and functional diversity, presumably to meet a variety of synaptic needs (*Sine and Engel, 2006*).

The archetypal pLGIC is the heteropentameric muscle-type nicotinic acetylcholine receptor (AChR), with the human adult form composed of two α-subunits, and one each of the β-, δ-, and ε-subunits (*Figure 1*; *Changeux, 2020a*; *Changeux, 2020b*). To gain insight into the structure, function, and evolution of the AChR, we have employed an ancestral reconstruction approach (*Emlaw et al., 2021*; *Emlaw et al., 2020*; *Prinston et al., 2017*; *Tessier et al., 2017*). Previously we resurrected a putative ancestral AChR β-subunit. Referred to as 'β$_{Anc}$', this subunit differed from its human counterpart by 132 amino acids (i.e. approximately 30% of the total amino acid sequence), yet was able to substitute for the human β-subunit, and also supplant the human δ-subunit, forming functional hybrid ancestral/human AChRs (*Emlaw et al., 2021*). These hybrid AChRs were ternary mixtures, containing two β$_{Anc}$ subunits, two human α-subunits, and one human ε-subunit. A concatameric construct confirmed that the two β$_{Anc}$ subunits resided next to each other, occupying positions in the heteropentameric complex usually reserved for the human β- and δ-subunits (*Figure 1*; *Emlaw et al., 2021*). Regardless of whether they bind agonist or not, all pLGIC subunits have both principal (+) and complementary (−) subunit interfaces. To sit next to each other, the (+) and (−) interfaces of any two neighbouring subunits must be structurally compatible. Thus, for β$_{Anc}$ to replace both the human β- and δ-subunits, the (+) and (−) interfaces of β$_{Anc}$ must be compatible with each other, raising the possibility that multiple β$_{Anc}$ subunits could coassemble to form β$_{Anc}$ homomers (*Figure 1*).

Here, using single-channel measurements, we demonstrate that β$_{Anc}$ readily forms homopentameric channels that open spontaneously in the absence of acetylcholine. This spontaneous activity displays hallmarks of the muscle-type AChR, including steady-state single-channel burst behaviour. These findings demonstrate how fundamental characteristics of AChR activation are independent of agonist, and not a result of the complex heteropentameric architecture of the muscle-type AChR. Finally, an alternate ancestral β-subunit, reconstructed using a phylogenetic tree that matched the accepted species relationships (*Emlaw et al., 2021*), and which only shared ~85% identity with β$_{Anc}$, revealed that these unexpected characteristics of β$_{Anc}$ are robust to phylogenetic uncertainty, and thus deeply embedded within AChR β-subunit structure and evolutionary history.

## Results

Our first inkling that β$_{Anc}$ may be able to form homomeric channels came from heterogeneity in single-channel recordings acquired after alterations to our original heterologous expression/transfection protocol. Typical cotransfection of cDNAs encoding human α-, δ-, and ε-subunits with a cDNA encoding β$_{Anc}$ leads to robust cell surface expression of β$_{Anc}$-containing hybrid AChRs (*Prinston et al., 2017*). In an attempt to lower overall AChR expression with the purpose of facilitating single-channel analysis, we reduced the total amount of subunit cDNA in our transfections (~sixfold), while maintaining the same 2:1:1:1 subunit cDNA ratio (by weight; α:β$_{Anc}$:δ:ε). Reducing the amount of cDNA lowered overall AChR expression as expected, but also led to heterogeneity in our patches (*Figure 2A*), which was not present in original single-channel recordings of β$_{Anc}$-containing AChRs (*Prinston et al., 2017*). Instead of a single population of channels with a uniform amplitude of ~10 pA, and a burst behaviour indicative of β$_{Anc}$-containing AChRs (*Figure 2A*; left inset), we also observed a second class of channels with a different kinetic signature, and an increased amplitude (*Figure 2A*; right inset). A similar trend was not observed when cDNA encoding the wild-type human β-subunit was cotransfected instead of β$_{Anc}$ (*Figure 2—figure*

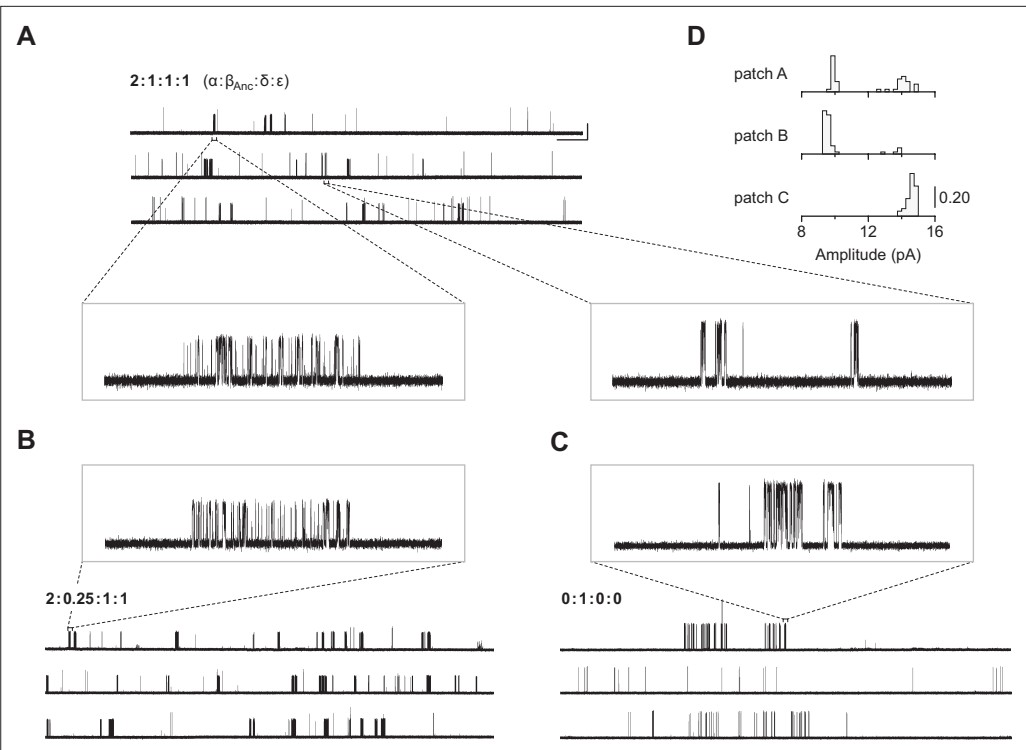

**Figure 2.** Single-channel recordings of $\beta_{Anc}$-containing channels. (**A**) Representative continuous recording from a cell-attached patch where cells were transfected with cDNAs encoding human muscle-type α-, δ-, and ε-subunits, and an additional cDNA encoding $\beta_{Anc}$ at a cDNA ratio of 2:1:1:1 (α:$\beta_{Anc}$:δ:ε). (**B**) Same as in (A), but where cells were transfected with an altered 2:0.25:1:1 cDNA ratio, making the $\beta_{Anc}$ subunit limiting, or (**C**) where cells were transfected with only the cDNA encoding $\beta_{Anc}$. In all cases openings are upward deflections, in the presence of 30 μM acetylcholine, and with an applied voltage of –120 mV. Continuous recordings are digitally filtered to 5 kHz, and the scale bar (2 s, 10 pA) in (A) applies to (B) and (C). Insets are digitally filtered to 10 kHz, with boxes representing scale bars (300 ms, 25 pA). (**D**) Event-based amplitude histograms for single-channel bursts from each of the patches shown in (A), (B), and (C). In each case, the height of the bins was normalised to the total number of bursts in each patch (A: 40; B: 50; C: 41), with the scale bar representing the indicated fraction (0.20) of the total bursts.

The online version of this article includes the following source data and figure supplement(s) for figure 2:

**Source data 1.** Unrasterized version of *Figure 2*.

**Figure supplement 1.** Single-channel recordings of the human adult muscle-type acetylcholine receptor exhibit homogeneous burst behaviour.

**Figure supplement 1—source data 1.** Unrasterized version of *Figure 2—figure supplement 1*.

---

*supplement 1*), indicating that $\beta_{Anc}$ was the source of the heterogeneity. Consistent with this, lowering the proportion of $\beta_{Anc}$ cDNA in the transfection mixture reduced the fraction of high amplitude channels (*Figure 2B and D*), while transfecting exclusively with $\beta_{Anc}$ cDNA resulted in patches where all channel openings had a uniformly high amplitude (*Figure 2C and D*). This demonstrated that when transfected alone, $\beta_{Anc}$ forms functional ion channels. The present work stems from this unexpected observation and describes characterisation of these previously unobserved channels.

The traces in *Figure 2* were recorded in the presence of agonist (30 μM acetylcholine). In the human adult muscle-type AChR, the agonist-binding sites are located at the α–δ and α–ε interfaces, and the β-subunit is the only subunit that does not participate directly in agonist binding (*Figure 1*; *Rahman et al., 2022*; *Rahman et al., 2020*; *Zarkadas et al., 2022*). We were therefore surprised to see single-channel activity in patches from cells transfected exclusively with $\beta_{Anc}$, as channels formed from a muscle-type β-subunit alone would not be expected to have intact agonist-binding sites. To determine if the activity of $\beta_{Anc}$-alone channels was dependent upon acetylcholine, we recorded

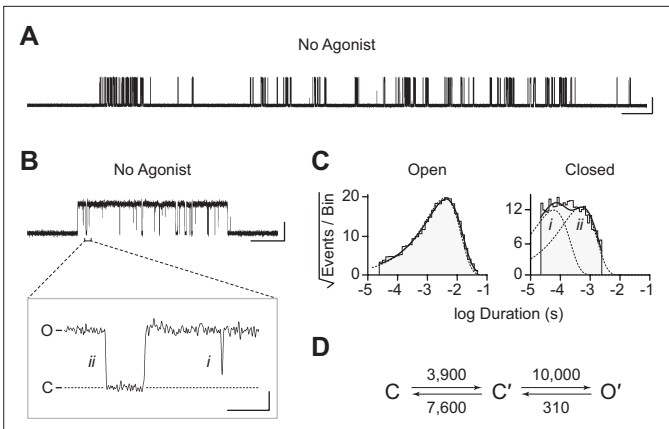

**Figure 3.** Spontaneous single-channel openings of β$_{Anc}$ homomers. (**A**) Representative continuous recording of a cell-attached patch from cells transfected with a single cDNA encoding β$_{Anc}$. Recording was made in the absence of acetylcholine and at an applied voltage of –120 mV. Data was digitally filtered to 5 kHz (scale bar = 2 s, 10 pA). (**B**) Single burst of openings from a homomeric β$_{Anc}$ channel, shown digitally filtered to 10 kHz (scale bar = 25 ms, 10 pA). Inset depicts (*i*) brief and (*ii*) long closings within bursts, where the former (*i*) are reminiscent of '*nachschlag* shuttings' (scale bar = 1 ms, 5 pA). (**C**) Open and closed dwell duration histograms for the representative patch depicted in (B). Individual exponential components determined manually (dashed lines) and kinetic fits from MIL (solid lines) are overlaid. Global kinetic fitting was performed on three individual recordings, from two separate transfections. (**D**) The single-channel data fit a three-state scheme (*Scheme 1*), where C, C', and O' correspond to closed, closed-primed, and open-primed states. Rate constants with units s$^{-1}$ are shown above and below corresponding arrow, with error estimates provided in *Table 1*.

The online version of this article includes the following source data and figure supplement(s) for figure 3:

**Source data 1.** Source data for *Figure 3*.

**Source data 2.** Unrasterized version of *Figure 3*.

**Figure supplement 1.** Kinetic fitting of alternate schemes describing spontaneous single-channel activity of β$_{Anc}$ homomers.

single-channel activity in the absence of acetylcholine (*Figure 3*). When no acetylcholine was present, patches from cells that were transfected exclusively with β$_{Anc}$ cDNA still displayed single-channel activity, indicating that β$_{Anc}$-alone channels open spontaneously under these conditions (*Figure 3A*). Furthermore, spontaneous activity occurred as bursts of closely spaced openings, separated by brief closings (*Figure 3B*; *Colquhoun and Hawkes, 1982*). The briefest of these intervening closings were reminiscent of classic '*nachschlag* shuttings' (*Figure 3B*, '*i*' in inset), observed in early patch clamp recordings from frog end-plate nicotinic receptors, and originally thought to relate to agonist efficacy (*Colquhoun and Sakmann, 1981*). Thus, despite being homomeric and lacking agonist-binding sites, β$_{Anc}$-alone channels display single-channel hallmarks of the muscle-type AChR.

To gain insight into the spontaneous activity of β$_{Anc}$-alone channels, we performed kinetic analysis of our single-channel data. First, we determined a critical closed duration ($\tau_{crit}$) to define bursts arising from a single ion channel. Then we determined the minimum number of components in our apparent open and closed dwell duration histograms by fitting each with a sum of exponentials. Open duration histograms were fit well by a single exponential component, while closed duration histograms required at least two components (*Figure 3C*). This suggested that a minimal scheme with a single open state and two closed states is necessary and sufficient to describe the spontaneous activity of β$_{Anc}$-alone channels. Based on this, we then fit the sequence of single-channel dwells using the three possible kinetic schemes, two linear and one cyclic, comprising a single open state and two closed states (*Figure 3D*; *Figure 3—figure supplement 1*). As a control, we also fit a simplified two-state scheme, where a single open state was connected to a single closed state, which, based on the relatively poor fit of the closed durations, confirmed that inclusion of a second closed state was justified (*Figure 3—figure supplement 1*). Overlaying the resulting fits on top of duration histograms revealed that each of the possible three-state schemes fits the observed dwells equally well, thus discriminating between the possible kinetic schemes is not trivial (*Figure 3—figure supplement 1*). We settled

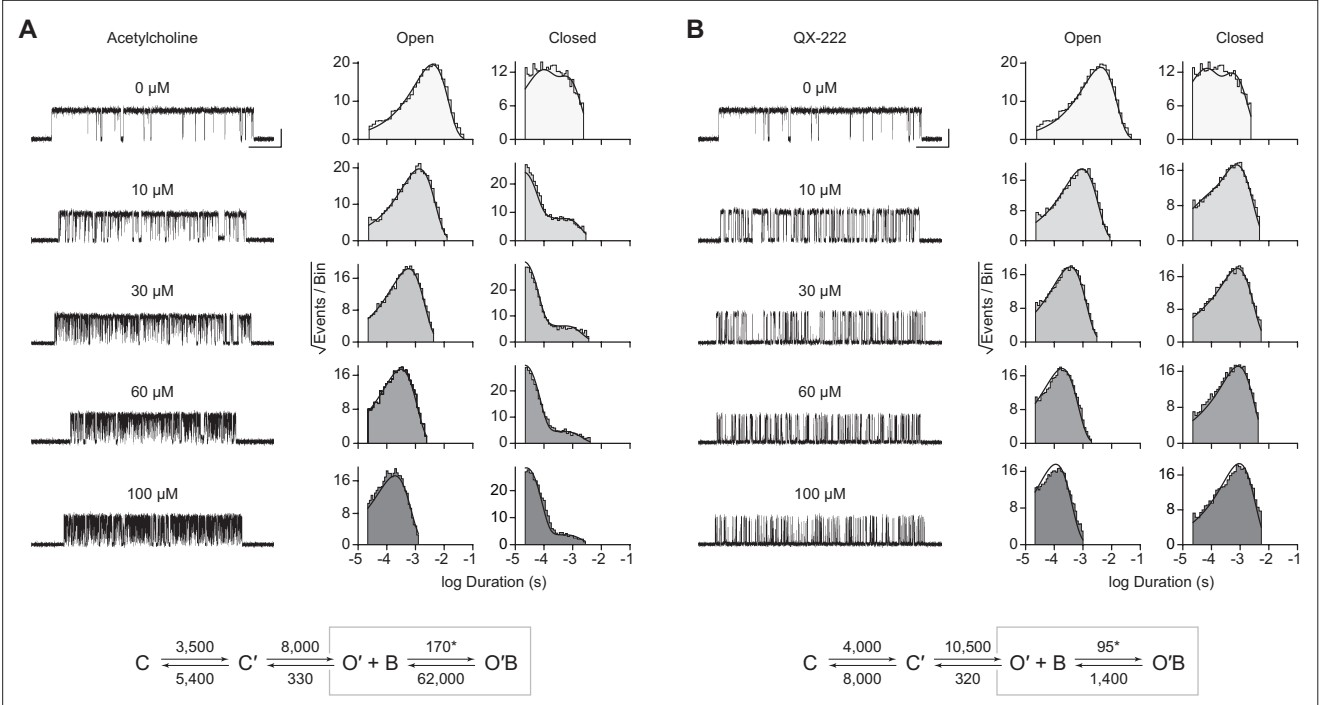

**Figure 4.** Open-channel block of $\beta_{Anc}$ homomers by acetylcholine and 2-[(2,6-dimethylphenyl)amino]-$N,N,N$-trimethyl-2-oxoethaniminium chloride (QX-222). Representative single-channel activity of $\beta_{Anc}$ homomers in the presence of increasing concentrations of (**A**) acetylcholine and (**B**) QX-222. Openings are upward deflections. Recordings were obtained with an applied voltage of –120 mV. Data were filtered to 10 kHz (scale bars = 25 ms, 10 pA; applies to (A) and (B)). The sequence of dwells from each dataset, encompassing the full concentration range of the blocker, was globally fit to the same three-state scheme used for $\beta_{Anc}$, where an additional fourth state corresponding to the open/blocked channel was added (**Scheme 2**). Global kinetic fits were performed on three individual recordings for each concentration of blocker, from at least two separate transfections, corresponding to 15 total patches for each global fit. Note that the recordings in the absence of blocker are the same for each dataset. Rate constants are overlaid on the scheme below each dataset, with error estimates presented in **Table 1**.

The online version of this article includes the following source data and figure supplement(s) for figure 4:

**Source data 1.** Source data for **Figure 4**.

**Source data 2.** Unrasterized version of **Figure 4**.

**Figure supplement 1.** Single-channel kinetics of the human adult muscle-type acetylcholine receptor (AChR).

**Figure supplement 1—source data 1.** Source data for **Figure 4—figure supplement 1**.

**Figure supplement 1—source data 2.** Unrasterized version of **Figure 4—figure supplement 1**.

**Figure supplement 2.** Kinetics of 2-[(2,6-dimethylphenyl)amino]-$N,N,N$-trimethyl-2-oxoethaniminium chloride (QX-222) block of the human adult muscle-type acetylcholine receptor.

**Figure supplement 2—source data 1.** Source data for **Figure 4—figure supplement 2**.

**Figure supplement 2—source data 2.** Unrasterized version of **Figure 4—figure supplement 2**.

upon the simple linear scheme, where $\beta_{Anc}$-alone channels transition from a closed state, C, to an intermediate closed state, C′, before opening to O′. The form of this scheme, with an intermediate closed state that precedes channel opening, is guided by models of AChR activation that include a single 'flipping' or multiple 'priming' steps (**Lape et al., 2008**; **Mukhtasimova et al., 2016**; **Mukhtasimova et al., 2009**). Given that for $\beta_{Anc}$-alone channels there is a single intermediate closed state that precedes channel opening, we refer to this state in our scheme as 'flipped' or 'primed'.

In the presence of acetylcholine, the single-channel current traces appeared different (**Figure 4A**). As the concentration of acetylcholine increased from 10 to 100 µM, there was a progressive decrease in the duration of openings, as well as a reciprocal increase in the number of short-lived closings within each burst (**Figure 4A**). This can be observed as a leftward shift in the open duration histograms as the concentration of acetylcholine is increased. This single-channel behaviour is a hallmark of open-channel block, a ubiquitous property of AChR agonists, including acetylcholine (**Lape et al., 2009**; **Mukhtasimova et al.,**

**Table 1.** Single-channel kinetics of spontaneously opening $\beta_{Anc}$ homomers.

| Homomer | $k_{+1}'$ | $k_{-1}'$ | $K'$ | $\beta_1$ | $\alpha_1$ | $\Theta_1$ | $k_{+B}$ | $k_{-B}$ | $K_B$ (μM) |
|---|---|---|---|---|---|---|---|---|---|
| No agonist (3 patches) | 3900 (110) | 7600 (450) | 0.51 | 10,000 (300) | 310 (3) | 32.26 | N/A | N/A | N/A |
| Acetylcholine (15 patches) | 3500 (85) | 5400 (265) | 0.65 | 8000 (200) | 330 (3) | 24.24 | 170* (1.8) | 62,000 (450) | 364.71 |
| QX-222 (15 patches) | 4000 (110) | 8000 (440) | 0.50 | 10,500 (300) | 320 (3) | 32.81 | 95* (0.5) | 1400 (7) | 14.74 |
| Acetylcholine (Constrained) (15 patches) | *3900* | *7600* | *0.51* | *10,000* | *310* | *32.26* | 170* (1.5) | 60,800 (410) | 357.65 |
| QX-222 (Constrained) (15 patches) | *3900* | *7600* | *0.51* | *10,000* | *310* | *32.26* | 95* (0.5) | 1400 (6.5) | 14.74 |

Note: Rate constants were estimated from fitting **Scheme 1** or **Scheme 2** presented in **Figures 3 and 4**, respectively. Data were globally fit (number of patches indicated in each case) over a range of acetylcholine/2-[(2,6-dimethylphenyl)amino]-*N,N,N*-trimethyl-2-oxoethaniminium chloride (QX-222) concentrations, with rate constants and associated errors (parentheses) estimated by MIL (see Materials and methods). Priming ($K'$), gating ($\theta_t$), and blocking ($K_B$) equilibrium constants represent $k_{+1}'/k_{-1}'$, $\beta_t/\alpha_t$, and $k_B/k_{+B}$, respectively. Association rate constants (*); ($k_{+B}$) are presented in units of μM$^{-1}$·s$^{-1}$, while remaining rate constants are presented in units of s$^{-1}$. Constrained rates (presented in italicized) were held constant at the values determined from the agonist-free dataset, while the blocking rate constants were estimated.

**2016**; **Ogden and Colquhoun, 1985**; **Sine and Steinbach, 1984**). Consistent with this blockage profile, the same trend, albeit with longer-lived blocking events, was observed with the well-characterized AChR open-channel blocker, 2-[(2,6-dimethylphenyl)amino]-*N,N,N*-trimethyl-2-oxoethaniminium chloride *(QX-222)* (**Figure 4B**; **Charnet et al., 1990**; **Leonard et al., 1988**; **Pascual and Karlin, 1998**).

The kinetics of open-channel block are determined by interactions between the blocking molecule and residues that line the channel pore in the open state, and therefore provide indirect structural insight into the open state. To compare the open state structures of $\beta_{Anc}$-alone homomers with wild-type AChRs, we determined the kinetics of acetylcholine and QX-222 block for both types of channels (**Figure 4**; **Figure 4—figure supplements 1 and 2**). To fit our $\beta_{Anc}$-alone single-channel data recorded in the presence of a blocker, we introduced an additional open, but blocked (i.e. non-conducting) state connected to our open state, where the forward rate of block was dependent upon the concentration of the blocking molecule (**Figure 4A and B**; **Neher and Steinbach, 1978**). We then globally fit each of our $\beta_{Anc}$-alone datasets encompassing between 0 and 100 μM acetylcholine or QX-222. Initially, we restricted the rates of the core (C-C′-O′) scheme to those inferred in the absence of blocker; however, allowing all parameters to be estimated, led to negligible changes in the inferred rates of block. For $\beta_{Anc}$-alone and wild-type channels, the rates of acetylcholine and QX-222 block were comparable (see **Table 1** and **Table 2**). Of note, while the forward rate of QX-222 block ($k_{+B}$) was almost the same for both types of channels, the reverse rate of QX-222 unblocking ($k_{-B}$) was nearly twofold faster for $\beta_{Anc}$-alone channels, indicating that the open pore of $\beta_{Anc}$-alone channels has a slightly reduced affinity for QX-222. Regardless of this nuance, the similarity in the profiles of acetylcholine and QX-222 block suggests that the structure of the open pore in the two types of channels is similar.

**Table 2.** Kinetics of acetylcholine (ACh) activation and 2-[(2,6-dimethylphenyl)amino]-*N,N,N*-trimethyl-2-oxoethaniminium chloride (QX-222) block of human adult muscle-type acetylcholine receptors (AChRs).

| WT | $k_{+1}$ | $k_{-1}$ | $K_1$ (μM) | $k_{+2}$ | $k_2$ | $K_2$ (μM) | $\beta_1$ | $\alpha_1$ | $\Theta_1$ | $\beta_2$ | $\alpha_2$ | $\Theta_2$ | $k_{+B\,(ACh)}$ | $k_{-B(ACh)}$ | $K_{B(ACh)}$ (μM) | $k_{+B}$ (QX-222) | $k_{-B}$ (QX-222) | $K_B$ (QX-222) (μM) |
|---|---|---|---|---|---|---|---|---|---|---|---|---|---|---|---|---|---|---|
| ACh (24 pat.) | 650* (35) | 14,400 (1000) | 22.12 | 325* (20) | 26,500 (400) | 81.54 | 33 (2.5) | 8750 (650) | 3.77 E-03 | 14,000 (450) | 1000 (10) | 14 | 215* (3) | 110,000 (800) | 511.63 | N/A | N/A | N/A |
| QX-222 (10μM ACh) (15 pat.) | *6500* | *14,400* | *2.21* | *3250* | *26,500* | *8.15* | *33* | *8750* | *3.77 E-03* | *14,000* | *1000* | *14* | *2150* | *110,000* | *51.16* | 100* (0.5) | 845 (5.5) | 8.45 |
| QX-222 (30μM ACh) (15 pat.) | *19,500* | *14,400* | *0.74* | *9750* | *26,500* | *2.72* | *33* | *8750* | *3.77 E-03* | *14,000* | *1000* | *14* | *6450* | *110,000* | *17.05* | 110* (0.5) | 845 (4.5) | 7.68 |

Note: Rate constants were estimated from fitting **Scheme 3** or **Scheme 4** presented in **Figure 4—figure supplements 1 and 2**, respectively. Where 'A' represents agonist, and 'R', 'R*', and 'R*$_B$' represent the closed, open, and open-blocked states of the human adult muscle-type acetylcholine receptor (AChR). Data were globally fit (number of patches indicated in each case) over a range of 2-[(2,6-dimethylphenyl)amino]-*N,N,N*-trimethyl-2-oxoethaniminium chloride (QX-222) concentrations with fixed concentration of acetylcholine (ACh: 10 μM or 30 μM), with rate constants and associated errors (parentheses) estimated with MIL (see Materials and methods). Apparent binding ($K_n$), apparent gating ($\theta_n$), and blocking ($K_B$) equilibrium constants represent $k_{-n}/k_{+n}$, $\beta_n/\alpha_n$, and $k_B/k_{+B}$, respectively. Association rate constants ($k_{+1}$, $k_{-2}$, and $k_{+B}$) are presented in units of μM$^{-1}$·s$^{-1}$, while remaining rate constants are presented in units of s$^{-1}$. Constrained rates (italicized) were held constant to the rates estimated from fitting of wild-type (WT) in the absence of QX-222, allowing the QX-222 blocking rates to be estimated independently.

Given that $\beta_{Anc}$-alone channels are expressed in the absence of other AChR subunits, a reasonable hypothesis is that they are homopentamers. To determine the subunit stoichiometry of $\beta_{Anc}$-alone channels, we employed a single-channel electrical fingerprinting strategy, where mutations altering unitary conductance were used to count the number of individual $\beta_{Anc}$ subunits in $\beta_{Anc}$-alone channels. A similar strategy has been employed with tetrameric potassium channels (**Niu and Magleby, 2002**), and other pLGICs (**Andersen et al., 2011**), including both the homopentameric $\alpha7$ AChR (**Andersen et al., 2013**; **daCosta et al., 2015**; **daCosta and Sine, 2013**) and the hetero-pentameric muscle-type AChR (**Emlaw et al., 2021**). The approach relies on identifying high-conductance (HC) and low-conductance (LC) variants of the $\beta_{Anc}$ subunit, and then co-expressing them to reveal a number of amplitude classes. Openings in each amplitude class originate from channels incorporating the same ratio of HC to LC subunits, and based on the total number of amplitude classes, the number of $\beta_{Anc}$ subunits within $\beta_{Anc}$-alone channels can be inferred.

When $\beta_{Anc}$ is expressed alone, the resulting channels exhibit a single, uniform amplitude, distributed around a mean of ~16 pA (**Figure 5A and D**), making the wild-type $\beta_{Anc}$ subunit an ideal HC subunit for electrical fingerprinting. To identify a LC variant of $\beta_{Anc}$, we took advantage of a structural feature inherent to eukaryotic pLGICs: as conducting ions exit the channel's transmembrane pore, they are obliged to pass through one of five portals in the cytoplasmic domain (**Rahman et al., 2020**; **Unwin, 2005**). Framed by charged or polar residues from each subunit, these portals influence single-channel conductance. Mimicking the homologous 5-HT$_{3A}$ receptor, which has an unusually low single-channel conductance (**Hales et al., 2006**; **Kelley et al., 2003**; **Peters et al., 2004**), we substituted three arginine residues (E420R, D424R, and E428R) into this region of $\beta_{Anc}$. When $\beta_{Anc}$ harbouring three arginines in this region was transfected by itself, the resulting channels exhibited a reduced single-channel amplitude centred around ~1–2pA (**Figure 5B and D**). With its markedly reduced amplitude, $\beta_{Anc}$ harbouring three arginine residues is a suitable LC subunit for electrical fingerprinting.

When cDNAs encoding HC and LC variants of $\beta_{Anc}$ were transfected together, a variety of single-channel amplitudes were observed in each patch (**Figure 5C**). The relative proportion of channels with high versus low amplitude could, to some degree, be tuned by the ratio of HC to LC $\beta_{Anc}$ cDNA used for transfection (**Figure 5—figure supplement 1**). Constructing event-based amplitude histograms, and pooling amplitudes from more than

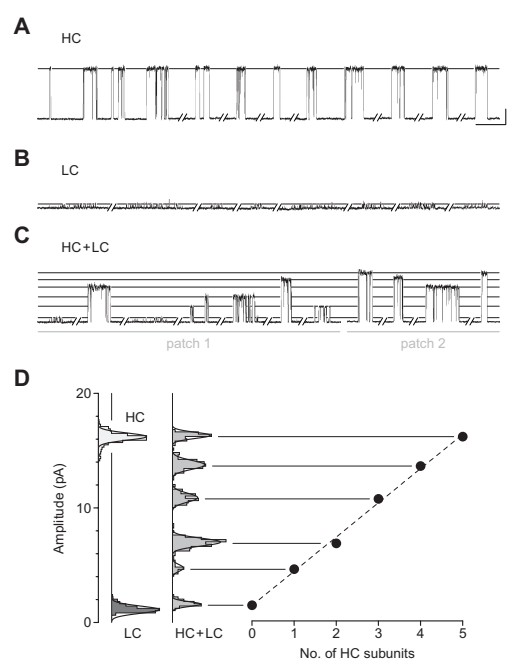

**Figure 5.** Electrical fingerprinting to determine subunit stoichiometry of $\beta_{Anc}$ homomers. Representative single-channel activity from cells transfected with (**A**) cDNA encoding the wild-type high-conductance (HC) $\beta_{Anc}$ subunit, or (**B**) a mutant low-conductance (LC) $\beta_{Anc}$ variant harbouring substitutions that reduce single-channel amplitude. (**C**) Cotransfection of cDNAs encoding HC and LC $\beta_{Anc}$ variants led to patches (two shown) with heterogeneous amplitudes. (**D**) The amplitudes segregate into six well-defined amplitude classes (total of 495 bursts combined from the two patches in (**C**)), where the highest and lowest amplitude classes match that of the all-HC (1569 bursts) and all-LC classes (883 bursts), respectively. Plot of the mean amplitude of each class as a function of the presumed number of incorporated HC subunits (error bars = standard deviations of the mean but are smaller than the points themselves). Recordings were obtained with an applied voltage of –120 mV, and traces were digitally filtered to 1 kHz to facilitate amplitude detection (scale bar = 50 ms, 5 pA; applies to (A), (B), and (C)).

The online version of this article includes the following source data and figure supplement(s) for figure 5:

**Source data 1.** Unrasterized version of **Figure 5**.

**Figure supplement 1.** Event-based amplitude histograms derived from a representative single patch where cells were cotransfected with cDNAs encoding for high-conductance (HC) and low-conductance (LC) variants of $\beta_{Anc}$ at a ratio (wt:wt) of (**A**) 1HC:4LC (208 total bursts) or (**B**) 4HC:1LC (287 total bursts).

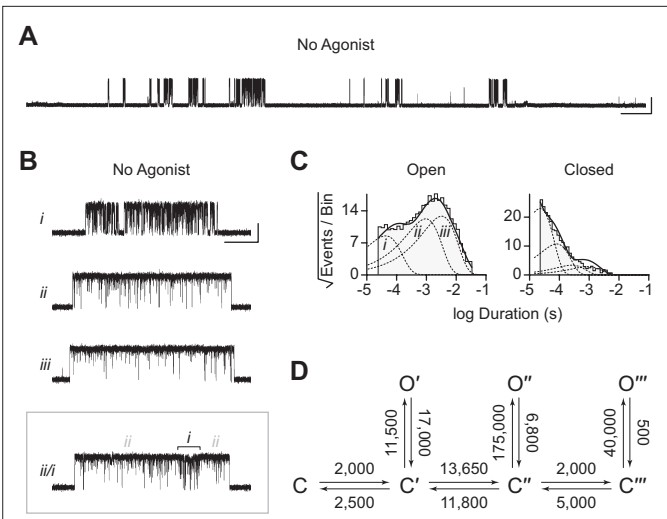

**Figure 6.** Spontaneous single-channel openings of homomers formed from an alternate ancestral β-subunit (β$_{AncS}$). (**A**) Representative continuous recording of a cell-attached patch from cells transfected with a single cDNA encoding β$_{AncS}$. Recording was made in the absence of acetylcholine, at an applied voltage of –120 mV, where spontaneous openings are upward deflections. Data was digitally filtered to 5 kHz (scale bar = 2 s, 10 pA). (**B**) Bursts from homomeric β$_{AncS}$ channels, each exhibiting one of three different types (*i*, *ii*, *iii*) of openings (scale bar = 25 ms, 10 pA). The boxed burst at the bottom is an example of a single burst that contains more than one type of opening (*ii/i*). (**C**) Open and closed dwell duration histograms for the representative patch depicted in (B). Individual exponential components determined manually (dashed lines) and kinetic fits from MIL (solid lines) are overlaid. Global kinetic fitting was performed on three individual recordings, from two separate transfections. The exponential components (*i*, *ii*, *iii*) in the open duration histogram correspond to the different types of openings observed within the bursts in panel (B). (**D**) The three-state scheme in *Figure 3* (*Scheme 1*) was expanded to include additional priming steps ('singly', 'doubly', and 'triply' primed), each with their own connected open state (*Scheme 5*). Rate constants are shown, with error estimates provided in *Table 3*.

The online version of this article includes the following source data for figure 6:

**Source data 1.** Source data for *Figure 6*.

**Source data 2.** Unrasterized version of *Figure 6*.

one recording, revealed that the amplitudes segregated into as many as six amplitude classes, with the highest and lowest amplitude classes matching that of the HC and LC forms of β$_{Anc}$-alone channels. The difference in amplitude between successive classes was somewhat regular, demonstrating five approximately equal contributions to single-channel conductance (*Figure 5D*), consistent with the hypothesis that β$_{Anc}$-alone channels are homopentamers.

As noted previously, reconstruction of β$_{Anc}$ was based upon a molecular phylogeny that diverged from the accepted species phylogeny (*Emlaw et al., 2021*; *Emlaw et al., 2020*; *Prinston et al., 2017*; *Tessier et al., 2017*). Reconstruction of ancestral protein sequences is based upon a best-fit model of amino acid evolution, a multiple sequence alignment, as well as a phylogenetic tree relating the sequences within the alignment (*Thornton, 2004*). We therefore wondered if the ability of β$_{Anc}$ to form spontaneously opening homomers was an artefact of the discordant tree used to reconstruct it. To test this, we took advantage of an alternate ancestral β-subunit, called 'β$_{AncS}$', whose reconstruction was based upon a molecular phylogeny that matched the accepted species phylogeny (*Emlaw et al., 2021*). Despite 67 substitutions and 6 indels relative to β$_{Anc}$, when expressed alone, β$_{AncS}$ still formed homomeric channels that opened in bursts in the absence of acetylcholine (*Figure 6*). This demonstrated that the ability of β$_{Anc}$ to form homomers that spontaneously open is not an artefact of the phylogeny used to reconstruct it. Instead, this surprising ability of β$_{Anc}$ is robust to the phylogenetic uncertainties inherent in ancestral sequence reconstruction, as well as substantial variation in the amino acid sequence of the reconstructed ancestral β-subunits.

While β$_{AncS}$ forms homomers that open spontaneously, inspection of the β$_{AncS}$ single-channel activity revealed additional complexity not seen with β$_{Anc}$. For β$_{AncS}$, single-channel bursts were heterogeneous, displaying at least three distinct kinetic behaviours (*Figure 6B*). In some cases, the kinetic behaviour

**Table 3.** Single-channel kinetics of spontaneously opening $\beta_{AncS}$ homomers.

| | $k_{+1}'$ | $k_{-1}'$ | $\beta_1$ | $\alpha_1$ | $\Theta_1$ | $k_{+2}''$ | $k_{-2}''$ | $\beta_2$ | $\alpha_2$ | $\Theta_2$ | $k_{+3}'''$ | $k_{-3}'''$ | $\beta_3$ | $\alpha_3$ | $\Theta_3$ |
|---|---|---|---|---|---|---|---|---|---|---|---|---|---|---|---|
| $\beta_{AncS}$ (3 pat.) | 2000 (80) | 2500 (140) | 11,500 (450) | 17,000 (800) | 0.676 | 13,650 (400) | 14,800 (2000) | 175,000 (17,000) | 6800 (1400) | 25.74 | 2000 (300) | 5000 (450) | 40,000 (1550) | 500 (15) | 80 |

Note: Rate constants were estimated from fitting **Scheme 5** in **Figure 6** (and below). Data were globally fit (three individual patches from two separate transfections with rate constants and associated errors (parentheses) estimated within MIL (see Materials and methods). Gating equilibrium ($\theta_n$) constants represent $\beta_n/\alpha_n$. Rate constants are presented as s$^{-1}$.

changed within a burst, demonstrating that the different kinetics were possible within the same channel (**Figure 6B**; boxed). This heterogeneity was also reflected in apparent open and closed duration histograms, with each displaying a minimum of three exponential components (**Figure 6C**). The increased number of exponential components indicated additional open and closed states relative to $\beta_{Anc}$, and thus that the three-state scheme used to fit $\beta_{Anc}$ was insufficient to describe the spontaneous activity of $\beta_{AncS}$. To account for the additional states, we expanded our original scheme to include two additional 'priming' steps, where openings could occur from one of three primed states (**Figure 6D**). This scheme, with multiple priming steps, builds directly upon the one used to fit muscle-type AChRs that had been engineered to open spontaneously (**Mukhtasimova et al., 2009**). Given the heterogeneity of the $\beta_{AncS}$ single-channel activity, and the complexity of this scheme, we caution against over interpretation of the inferred rates. Nevertheless, we note that in accord with the muscle-type AChR, the equilibrium gating constants appear to increase (**Table 3**; compare $\Theta_1$, $\Theta_2$, and $\Theta_3$), and thus the open states become more and more favoured, for each successive priming step. In any case, the fits suggest that a scheme of this form, with multiple stages of priming, is adequate to describe the complex spontaneous single-channel activity of $\beta_{AncS}$ homomers.

## Discussion

The goal of this work was to identify the source of heterogeneity in single-channel recordings from cells transfected with our previously described, reconstructed ancestral AChR β-subunit (**Prinston et al., 2017**). The heterogeneity in single-channel activity was only evident upon modification of our original transfection protocol. Specifically, when the total amount of transfected AChR subunit cDNA was reduced ~sixfold, a second class of single-channel openings with increased amplitude and a different kinetic signature was observed. Evidently, $\beta_{Anc}$ can both incorporate into heteropentameric AChRs (**Emlaw et al., 2021**), and self-associate to form homopentameric $\beta_{Anc}$ channels. Reducing the overall expression of AChR subunits was sufficient to tip the balance between the incorporation of $\beta_{Anc}$ into hetero- versus homopentamers, and thus unmask this previously unobserved ability of $\beta_{Anc}$ to form homomeric channels.

At first glance, the ability of $\beta_{Anc}$ to form homopentamers is unexpected. Modern-day muscle-type β-subunits do not appear to form homopentamers, and instead appear fully entrenched within heteropentameric muscle-type AChRs. Since reconstruction of $\beta_{Anc}$ was informed almost exclusively by modern muscle-type β-subunits, there was no reason to expect that $\beta_{Anc}$ would behave differently (**Prinston et al., 2017**). However, as mentioned previously, $\beta_{Anc}$ can replace both the human muscle-type β- and δ-subunits in hybrid ancestral/human AChRs (**Emlaw et al., 2021**). For this to be possible the principal (+) and complementary (−) subunit interfaces of $\beta_{Anc}$ must be compatible with each other (**Figure 1**), leading to the logical hypothesis, confirmed here, that $\beta_{Anc}$ can form homopentamers. From an evolutionary perspective, this ability to revert the muscle-type β-subunit, which is entrenched in a heteropentamer, back to a subunit capable of forming homopentamers, attests to the presumed homopentameric origins of pentameric ligand-gated ion channel subunits (**Ortells and Lunt, 1995**).

We have also shown that $\beta_{Anc}$ homopentamers open spontaneously. This is surprising. What is even more surprising is that the spontaneous single-channel activity of $\beta_{Anc}$ homopentamers resembles that of the agonist-activated muscle-type AChR. Some of the first single-channel recordings of frog AChRs activated by agonists revealed that agonist-activated openings occur in quick succession, as bursts of activity originating from the same channel (**Sakmann et al., 1980**). The main effect of increasing the concentration of agonist was to increase the open probability within bursts, with saturating concentrations of full agonists, such as acetylcholine, leading to bursts where the open probability exceeded 0.90 (i.e. the channel was open for more than 90% of the duration of the burst). Spontaneous openings of $\beta_{Anc}$ homopentamers also

occur in bursts, and despite the absence of agonist, activation appears efficient, with the probability of being open within a burst also exceeding 0.90. The kinetic structure of bursts of spontaneous $\beta_{Anc}$ openings also resembles that of the agonist-activated AChR, with bursts containing several types of closings, the briefest of which are reminiscent of classic 'nachschlag shuttings'. This is significant because it was originally proposed that the duration of nachschlag shuttings was related to agonist efficacy (**Colquhoun and Sakmann, 1981**). However, subsequent work showed that nachschlag shuttings were independent of agonist, which necessitated the introduction of an additional closed state appended to the agonist-activated open state in early schemes of AChR activation (**Sine and Steinbach, 1986**). This latter finding was some of the impetus for refined schemes, where the additional closed states preceded channel opening and were referred to as 'flipped' or 'primed' (**Lape et al., 2008**; **Mukhtasimova et al., 2009**). Our kinetic analysis has shown that the spontaneous activity of $\beta_{Anc}$ fits an analogous scheme, containing an intermediate closed state that also precedes channel opening, but where the agonist binding steps have been omitted due to the absence of agonist. Evidently, these functional hallmarks of AChR activation do not arise from the complex heteropentameric architecture of the muscle-type receptor, nor do they depend upon the presence of agonist. Instead, they are fundamental properties preserved and encoded in the reconstructed ancestral amino acid sequence of $\beta_{Anc}$ (and $\beta_{AncS}$).

Relative to the human β-subunit, which does not form spontaneously opening homopentamers, $\beta_{Anc}$ contains 125 individual substitutions and 7 deletions (**Prinston et al., 2017**). These 132 amino acid differences are scattered throughout the entire β-subunit, and presumably not all of them are required for $\beta_{Anc}$ to form spontaneously opening homopentamers. Thus, although the differences in amino acid sequence between $\beta_{Anc}$ and the human β-subunit are *sufficient* to convert the human β-subunit into a subunit capable of (1) forming homopentamers and (2) spontaneously opening, the large number of residues involved, as well as their delocalised nature, obscures the subset of residues strictly *necessary* for either (or both) functions. Nevertheless, several of the residues substituted between the human β-subunit and $\beta_{Anc}$ align with known determinants of homomeric assembly and spontaneous activity in various pLGICs (**Appendix 1—figure 1**).

Compared to the human wild-type and $\beta_{Anc}$-containing AChRs, $\beta_{Anc}$ homopentamers exhibit increased single-channel amplitude and thus conductance (**Figure 2D**; **Figure 5**; **Figure 2—figure supplement 1**). This increased conductance can be explained by amino acid differences in the second transmembrane segment (i.e. M2) of each subunit, which is one of the more conserved regions across all pLGICs. M2 lines the channel pore at its narrowest constriction and is a well-known determinant of single-channel conductance (**Imoto et al., 1986**). With exception of the 2' and 6' positions, M2 is conserved between $\beta_{Anc}$ and the human β-subunit (**Emlaw et al., 2020**). However, to reconcile the differences in conductance, it is important to consider the M2 segments of not just the β-subunits, but all the subunits that make up the various channels. Owing to the symmetric (homopentamers) or pseudosymmetric (heteropentamers) architecture of pLGICs, analogous M2 residues from each subunit occupy similar positions in the pore, forming concentric rings of amino acids. Several of these rings influence conductance, with the –4', –1', 2', and 20' positions being particularly important, and referred to as the 'cytoplasmic', 'intermediate', 'central', and 'extracellular' rings, respectively (**Imoto et al., 1991**; **Imoto et al., 1988**). In $\beta_{Anc}$ homopentamers, with five identical $\beta_{Anc}$ subunits, the amino acids contributed from individual subunits are identical. This is significant because $\beta_{Anc}$ contains acidic residues at the –4' (Asp), –1' (Glu), and 20' (Asp) positions, meaning that all 15 residues forming these three decisive rings are acidic, effectively maximising their negative potential (**Appendix 1—figure 2**). In heteropentamers, such as human wild-type and $\beta_{Anc}$-containing AChRs, the residues contributed by individual subunits are in some cases different, with polar/neutral (Gln) and basic (Lys) residues contributed by the ε- and δ-subunits, respectively (**Appendix 1—figure 2**). Substitution of neutral or basic residues into any of these three rings has been shown to reduce conductance (**Imoto et al., 1988**). Thus, unlike for $\beta_{Anc}$ homopentamers that have higher conductance, the negative potential of these rings is not maximised in human wild-type and $\beta_{Anc}$-containing heteropentamers.

Based upon the rates of open-channel block by acetylcholine and the well-known open-channel blocker, QX-222, the structure of the open pore in $\beta_{Anc}$ homopentamers and the wild-type AChR appear to be similar. Nevertheless, kinetic fitting revealed that compared to wild-type, $\beta_{Anc}$ homopentamers exhibit an almost twofold reduced affinity for QX-222. While we have employed the simplest model for open-channel block (**Neher and Steinbach, 1978**), which neglects potential increased complexity at higher QX-222 concentrations (**Neher, 1983**), the simplest interpretation is that, like conductance,

this reduced apparent affinity can be explained by amino acid differences in M2. Specifically, the 6′ and 10′ sites are known determinants of QX-222 block (*Charnet et al., 1990*; *Leonard et al., 1988*; *Pascual and Karlin, 1998*), with a more polar environment at 6′, and a less polar environment at 10′, favouring interaction with QX-222 (*Charnet et al., 1990*). In the wild-type AChR, the polarity of the 6′ ring is limited by a hydrophobic phenylalanine residue contributed by the human β-subunit (*Appendix 1—figure 2*). In $\beta_{Anc}$ homopentamers, the 6′ ring does not contain this phenylalanine residue and is instead composed of five polar serine residues, predicting an increased affinity for QX-222. Conversely, the 10′ ring in $\beta_{Anc}$ homopentamers contains five threonine residues making it more polar, and thus a less favourable environment for QX-222, than in the wild-type AChR, which contains two nonpolar alanine residues (from the δ- and ε-subunits) (*Appendix 1—figure 2*). Evidently, the energetic contributions of these two rings to QX-222 blockage balance out such that $\beta_{Anc}$ homopentamers exhibit only a slightly reduced apparent affinity for QX-222 compared to the wild-type AChR.

Phylogenetic analyses of pLGICs suggests that present-day heteropentameric subunits, such as those entrenched in the muscle-type AChR, evolved from ancestral subunits capable of self-associating to form homopentamers (*Dent, 2010*; *Jaiteh et al., 2016*; *Le Novère and Changeux, 1995*; *Ortells, 2016*; *Ortells and Lunt, 1995*; *Pedersen et al., 2019*; *Tsunoyama and Gojobori, 1998*). Within this framework, strictly entrenched subunits have lost compatibility between their (+) and (−) interfaces, and thus the ability to self-associate. This is significant because the pentameric architecture of pLGICs places a constraint upon when the ability to self-associate became dispensable, and thus when strict subunit entrenchment could have evolved. For the simplest heteropentamers, made up of two types of subunits, at least one of the subunit types will have to occupy a minimum of three positions within the pentamer. In these cases, two copies of the same subunit must sit next to each other, and thus retain the ability to self-associate. Indeed, in neuronal α4β2 heteropentameric AChRs, both the α4- and β2-subunits have retained the ability to self-associate and form α4–α4 or β2–β2 interfaces, meaning that in principle, a variety of subunit stoichiometries and arrangements are possible. While additional factors such as subunit packing, relative expression, trafficking and assembly, and the presence of cellular chaperones provide additional constraints that limit the stoichiometries and arrangements observed (*Mazzaferro et al., 2021*; *Moroni et al., 2006*; *Nelson et al., 2003*; *Son et al., 2009*; *Walsh et al., 2018*), the requirement for self-association must remain for at least one of the subunit types. Thus, while the ability to self-associate is not sufficient to permit all possible heteropentameric stoichiometries and arrangements, it is necessary.

Unlike neuronal heteropentameric channels, such as α4β2 above, whose subunit stoichiometries and arrangements exhibit a degree of plasticity, the subunits in the muscle-type AChR are strictly entrenched. In this case, the definition of *strict* entrenchment neglects the γ- for ε-subunit swap that occurs during development (*Mishina et al., 1986*), because it occurs at the same location within the heteropentamer, and thus does not alter the stoichiometry and arrangement of the α-, β-, or δ-subunits. The above structural considerations imply that such strict entrenchment should only be expected in heteropentamers composed of three or more subunit types, and where each of the subunits has also lost the ability to self-associate. Muscle-type AChRs fulfill both criteria, in that they are made up of four types of subunits that are each incapable of self-associating and forming homopentamers. The finding that reconstructed ancestral β-subunits retain the ability to self-associate and form homomers, suggests that, unlike their extant human and *Torpedo* counterparts, the ancestral β-subunits were not strictly entrenched, and that their entrenchment occurred in parallel lineages after the last common ancestor shared by humans and Torpediniformes (i.e. Gnathostomata).

A degree of uncertainty is inherent in the reconstruction of any ancestral protein, and to solidify evolutionary conclusions it is important to assess whether the functions of putative ancestors are robust to these uncertainties. We have shown that an alternate ancestral β-subunit ($\beta_{AncS}$), with 67 substitutions and 6 indels relative to $\beta_{Anc}$, was still able to form homomers that opened spontaneously. Furthermore, spontaneous openings of $\beta_{AncS}$ homomers also occurred in bursts that contained brief closings (i.e. '*nachschlag* shuttings') and had an open probability that exceeded 0.90. These findings demonstrate that the ability of $\beta_{Anc}$ to form spontaneously opening homomers is robust to substantial variations in the inferred ancestral β-subunit amino acid sequence. Thus, the capacity of these reconstructed ancestral β-subunits to form spontaneously opening homomers is deeply embedded in their structure and evolutionary history.

The kinetic behaviour of $\beta_{AncS}$ homomers was more complex than originally observed with $\beta_{Anc}$. This increased complexity necessitated the expansion of the original scheme used to fit $\beta_{Anc}$ to include two

additional priming steps. Both the original $\beta_{Anc}$ and expanded $\beta_{AncS}$ schemes parallel the scheme used to describe the muscle-type AChR, which included two priming steps (i.e. 'singly' and 'doubly' primed) that each correlated with conformational changes around the two agonist-binding sites (*Mukhtasimova et al., 2009*). In the case of $\beta_{AncS}$, which lacks agonist-binding sites, the simplest interpretation is that the different levels of priming correlate with conformational changes occurring within individual subunits. Although the states in *Scheme 5* are labelled as 'singly', 'doubly', and 'triply' primed, the three priming steps could represent the three most terminal priming steps, where (assuming that $\beta_{AncS}$ also forms homopentamers) three, four, or five $\beta_{AncS}$ subunits are primed. Within this framework, openings from $\beta_{AncS}$ channels with zero, one, or two primed subunits are presumably unstable, and thus not observed in our single-channel recordings. In an alternate scenario, 'singly' and 'doubly' primed could refer to $\beta_{AncS}$ channels with one or two primed subunits, respectively. While 'triply' primed could refer to channels with three or more primed subunits, but where openings from $\beta_{AncS}$ channels with three, four, or five primed subunits are indistinguishable. Applied to $\beta_{Anc}$, these interpretations suggest that either (1) openings from only the terminal priming step, where all five $\beta_{Anc}$ subunits are primed, are stable enough to be observed in $\beta_{Anc}$ homopentamers, or (2) openings can occur with fewer than five primed subunits, but these openings are kinetically indistinguishable. Regardless of the interpretation, priming is an important step in the activation of both $\beta_{AncS}$ and $\beta_{Anc}$ homopentamers.

Modern mechanisms of AChR activation include intermediate closed states, which place the roots of agonism at a stage in the activation process that precedes channel opening (*Lape et al., 2008*; *Mukhtasimova et al., 2016*; *Mukhtasimova et al., 2009*). A consequence of these mechanisms is that the ultimate opening and closing rates of the channel are independent of agonist. Here, we have shown that additional single-channel hallmarks of AChR function are independent of agonist, as they occur in spontaneously opening homopentameric channels that are devoid of agonist-binding sites, and which are formed from reconstructed ancestral AChR β-subunits. Often overlooked, the β-subunit is the least conserved of the four AChR subunits, and is the only subunit that does not contribute residues to the two AChR agonist-binding sites. Despite these considerations, hallmarks of AChR function remain deeply embedded in β-subunit sequence, structure, and evolutionary history. Given that these functional hallmarks are independent of agonist, it is tempting to speculate that they predate agonism, and thus that agonism evolved subsequently as an additional layer of regulation in this family of pentameric ion channels.

# Materials and methods

## Key resources table

| Reagent type (species) or resource | Designation | Source or reference | Identifiers | Additional information |
|---|---|---|---|---|
| Cell line (*Homo sapiens*) | BOSC 23 | ATCC | CRL11270 (discontinued) | Modified *Homo sapiens* embryonic kidney cells |
| Recombinant DNA reagent | pRBG4 – AChR α1 | Provided by Steven M. Sine (Mayo Clinic) | | *Homo sapiens* CHRNA1 (Accession: NM_000079.4) |
| Recombinant DNA reagent | pRBG4 – AChR β1 | Provided by Steven M. Sine (Mayo Clinic) | | *Homo sapiens* CHRNB1 (Accession: NM_000747.3) |
| Recombinant DNA reagent | pRBG4 – AChR δ | Provided by Steven M. Sine (Mayo Clinic) | | *Homo sapiens* CHRND (Accession: NM_000751.3) |
| Recombinant DNA reagent | pRBG4 – AChR ε | Provided by Steven M. Sine (Mayo Clinic) | | *Homo sapiens* CHRNE (Accession: NM_000080.4) |
| Recombinant DNA reagent | pRBG4 – AChR $\beta_{Anc}$ | Custom gene synthesis | | Construct originating from: PMID:28689969 |
| Recombinant DNA reagent | pRBG4 – AChR $\beta_{AncS}$ | Custom gene synthesis | | Construct originating from: PMID:33579823 |
| Recombinant DNA reagent | pGreenLantern | Provided by Steven M. Sine (Mayo Clinic) | | |
| Commercial assay or kit | Q5 DNA polymerase | New England Biolabs, inc | M0491 | PCR |
| Sequence-based reagent | SDM_AChRBAncLC_F | This paper | Mutagenesis primer | AGAACGCTGAAGAGAGACTGGCAGTACGTGGCCAT |
| Sequence-based reagent | SDM_AChRBAncLC_R | This paper | Mutagenesis primer | ATAGTCCTCTCTTTTCTGCAGCTGCTCAGCGAT |
| Chemical compound, drug | Acetylcholine Chloride | Sigma | A9101-10VL | Purity: 99% |
| Chemical compound, drug | QX-222 | Tocris | 1043/10 | Purity:>98% |
| Software, algorithm | TAC 4.3.3 | Bruxton (https://www.bruxton.com/legacy.html) | | Single-channel recording, detection, and analysis |

*Continued on next page*

*Continued*

| Reagent type (species) or resource | Designation | Source or reference | Identifiers | Additional information |
|---|---|---|---|---|
| Software, algorithm | R | https://www.r-project.org/ | | Open-source statistical computing software |
| Software, algorithm | scbursts | https://cran.r-project.org/web/packages/scbursts/index.html | | R Package – single-channel burst analysis |
| Software, algorithm | extreme-values | https://cran.r-project.org/web/packages/extremevalues/index.html | | R Package – outlier detection |
| Software, algorithm | MASS | https://cran.r-project.org/web/packages/MASS/index.html | | R Package – function and statistical analysis |
| Software, algorithm | xlsx | https://cran.r-project.org/web/packages/xlsx/index.html | | R Package – read and write excel files |

## Materials

The 2-[(2,6-dimethylphenyl)amino]-*N,N,N*-trimethyl-2-oxoethaniminium chloride (QX-222) was purchased from Tocris Bioscience. All other chemicals, including acetylcholine chloride, were purchased from Sigma-Aldrich.

## Molecular Biology

cDNAs of human muscle-type AChR subunits ($\alpha1$, $\beta1$, $\delta$, and $\varepsilon$) in the pRBG4 plasmid were provided by Steven M. Sine (Mayo Clinic), while cDNAs encoding $\beta_{Anc}$ and $\beta_{AncS}$ were reconstructed and cloned into pRBG4 as described previously (*Emlaw et al., 2021*; *Prinston et al., 2017*). Mutations to produce the LC variant of $\beta_{Anc}$ (E420R, D424R, E428R) were introduced by inverse PCR (*Silva et al., 2017*). Sanger sequencing confirmed the entire reading frame for all constructs.

## Mammalian cell expression

Combinations of human and ancestral subunit cDNAs were transfected into BOSC 23 cells (*Pear et al., 1993*), originally from ATCC (CRL11270), but provided by Steven M. Sine (Mayo Clinic) (RRID:CVCL_4401). Cells were maintained in Dulbecco's modified Eagle's medium (DMEM; Corning) containing 10% (vol/vol) fetal bovine serum (Gibco) at 37°C, until they reached 50–70% confluency. Cells were then transfected using calcium phosphate precipitation, and transfections terminated after 3–4 h by exchanging the medium. All experiments were performed one day post transfection (between 16 and 24 h after exchanging the medium). A separate plasmid encoding green fluorescent protein was included in all transfections to facilitate identification of transfected cells.

## Cell line authentication and mycoplasma testing

Approximately 5 million confluent cells were harvested and their total DNA isolated (E.Z.N.A. Tissue DNA Kit), and then submitted to The Centre for Applied Genomics Genetic Analysis Facility (The Hospital for Sick Children, Toronto, Canada) for STR profiling using Promega's GenePrint 24 System. A similarity search on the 8,159 human cell lines with STR profiles in Cellosaurus release 42.0 was conducted on the resulting STR profile, which revealed that the cell line shares closest identity (88%, CLASTR 1.4.4 STR Similarity Search Tool score) with cell line Anjou 65 (CVCL_3645). Anjou 65 is a child of CVCL_1926 (HEK293T/17) and is itself a parent line of CVCL_X852 (Bartlett 96). Bartlett 96 is the parent line of BOSC 23 (*Pear et al., 1993*). PCR tests confirmed that the cells were free from detectable mycoplasma contamination (Uphoff and Drexler, 2011, 2002).

## Single-channel patch clamp recordings

Single-channel patch clamp recordings were performed as previously described (*Mukhtasimova et al., 2016*). Recordings from BOSC 23 cells transiently transfected with cDNAs encoding wild-type, ancestral, or LC subunits, were obtained in a cell-attached patch configuration. All recordings were obtained with a membrane potential of –120 mV, with room temperature maintained between 20 and 22°C. The external bath solution contained 142 mM KCl, 5.4 mM NaCl, 0.2 mM CaCl$_2$, and 10mM 4-(2-hydroxyethyl)–1-piperazineethanesulfonic acid (HEPES), adjusted to pH 7.40 with KOH. The pipette solution contained 80 mM KF, 20 mM KCl, 40 mM K-aspartate, 2 mM MgCl$_2$, 1 mM ethylene glycol-bis($\beta$-aminoethyl ether)-*N,N,N',N'*-tetraacetic acid, and 10 mM HEPES, adjusted to a pH of 7.40 with

KOH. Acetylcholine and QX-222 were added to pipette solutions at their desired final concentrations and stored at –80°C. Patch pipettes were fabricated from type 7052 or 8250 non-filamented glass (King Precision Glass) with inner and outer diameters of 1.15 and 1.65 mm, respectively, and coated with SYLGARD 184 (Dow Corning). Prior to recording, electrodes were heat polished to yield a resistance of 5–8 MΩ. Single-channel currents were recorded on an Axopatch 200B patch clamp amplifier (Molecular Devices), with a gain of 100 mV/pA and an internal Bessel filter at 100 kHz. Data were sampled at 1.0 µs intervals using a BNC-2090 A/D converter with a National Instruments PCI 6111e acquisition card and recorded by the program Acquire (Bruxton).

### Dwell time and kinetic analysis

Single-channel detections were performed using the program TAC 4.3.3 (Bruxton). Data were analysed with an applied 10 kHz digital Gaussian filter. Opening and closing transitions were detected using the 50% threshold crossing criterion, and open and closed dwell duration histograms were generated within the program TACfit 4.3.3 (Bruxton). Histograms were visually fit with a minimum sum of exponential components. From the closed duration histograms, the intersection of the slowest activation and fastest inactivation/desensitisation components was taken as the critical closed duration ($\tau_{crit}$) (*Colquhoun and Hawkes, 1982*; *Sine et al., 1990*). Closings longer than $\tau_{crit}$ (corresponding to inter-burst closings) were removed from analysis. Events were imported into R using *scbursts* (*Drummond et al., 2019*), where individual durations were corrected for instrument risetime (*Colquhoun and Sigworth, 1995*), and bursts were defined by a $\tau_{crit}$ (*Sine et al., 1990*). Bursts with fewer than three events were omitted from further analysis. The probability of being open within a burst (i.e. burst $P_{open}$) was calculated for each burst, and bursts with a $P_{open}$ that did not fit within the normal distribution were removed using *extremevalues* (*van der Loo, 2010*). The distribution of burst $P_{open}$ was then fit with a Gaussian distribution, and bursts within two standard deviations from the mean were used for further kinetic analysis (*Drummond et al., 2019*). A user-defined kinetic scheme (see *Figure 3—figure supplement 1* for $\beta_{Anc}$, and the modified del Castillo and Katz scheme (*Colquhoun and Sakmann, 1985*) for the human adult AChR in *Figure 4—figure supplement 1*) was fit to the sequence of single-channel dwells in the global dataset using maximum likelihood implemented within MIL (QUB suite, State University of New York, Buffalo, NY). With a user-defined dead time of 18.83 µs, MIL corrected for missed events, estimated model parameters by maximum likelihood, and gave standard errors of the estimated parameters (see *Tables 1–3*; *Qin et al., 1996*).

### Electrical fingerprinting

The HC and LC variants of $\beta_{Anc}$ were transfected at 1:4 and 4:1 (HC:LC) cDNA ratios. Transfections and single-channel recordings were performed as described above. For detections, data were filtered to 1 kHz, and bursts defined by a uniform $\tau_{crit}$ of 2 ms imposed upon all recordings. Using the program TAC 4.3.3 (Bruxton), amplitudes of single-channel bursts were measured as the difference between open- and closed-channel currents. Amplitudes of individual bursts were pooled from separate recordings to generate event-based amplitude histograms (EBAHs; *Figure 5*), which were fit with Gaussian distributions within TACfit (Bruxton).

### Schemes

$$C \underset{k_{-1}{'}}{\overset{k_{+1}{'}}{\rightleftharpoons}} C' \underset{\alpha_1}{\overset{\beta_1}{\rightleftharpoons}} O'$$

**Scheme 1.** Kinetic scheme describing spontaneous single-channel activity of $\beta_{Anc}$ homomers.

$$C \underset{k_{-1}{'}}{\overset{k_{+1}{'}}{\rightleftharpoons}} C' \underset{\alpha_1}{\overset{\beta_1}{\rightleftharpoons}} O' + B \underset{k_{-B}}{\overset{k_{+B}}{\rightleftharpoons}} O'B$$

**Scheme 2.** Kinetic scheme describing single-channel activity of $\beta_{Anc}$ homomers in the presence of acetylcholine or 2-[(2,6-dimethylphenyl)amino]-N,N,N-trimethyl-2-oxoethaniminium chloride (QX-222).

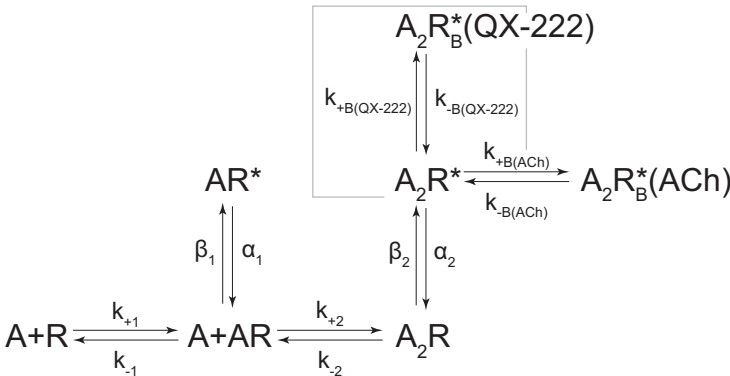

**Scheme 3.** Kinetic scheme describing single-channel activity of the human adult muscle-type acetylcholine (ACh) receptor.

**Scheme 4.** Kinetic scheme describing single-channel activity of the human adult muscle-type acetylcholine (ACh) receptor in the presence of 2-[(2,6-dimethylphenyl)amino]-N,N,N-trimethyl-2-oxoethaniminium chloride (QX-222).

**Scheme 5.** Kinetic scheme describing spontaneous single-channel activity of $\beta_{AncS}$ homomers.

## Acknowledgements

CJGT was funded in part by an Ontario Graduate Scholarship, while JRE was funded in part by a Natural Sciences and Engineering Research Council of Canada (NSERC) CREATE Scholarship. CJBdC acknowledges grants from NSERC (RGPIN-2016–04801), the Canada Foundation for Innovation (34475), the New Frontiers in Research Fund (NFRFE-2018–453 00064), and the Canadian Institutes of Health Research (377068).

## Additional information

### Funding

| Funder | Grant reference number | Author |
| --- | --- | --- |
| Natural Sciences and Engineering Research Council of Canada | Discovery Grant RGPIN-2016-04801 | Corrie JB daCosta |

| Funder | Grant reference number | Author |
|---|---|---|
| Canadian Institutes of Health Research | Project Grant 377068 | Corrie JB daCosta |
| Ontario Council on Graduate Studies, Council of Ontario Universities | Graduate Scholarship | Christian JG Tessier |
| New Frontiers in Research Fund | NFRFE-2018-453 00064 | Corrie JB daCosta |

The funders had no role in study design, data collection and interpretation, or the decision to submit the work for publication.

## Author contributions

Christian JG Tessier, CJGT acquired and analyzed all electrophysiological data, except the data for the electrical fingerprinting, which was recorded and analyzed by RMS, CJGT, and CJBdC. CJGT and CJBdC interpreted the data and wrote the manuscript., Conceptualization, Formal analysis, Investigation, Writing – original draft, Writing – review and editing, Methodology; Raymond M Sturgeon, Conceptualization, Formal analysis, Investigation, Methodology, RMS recorded and analyzed electrophysiological data for the electrical fingerprinting experiments., Writing – original draft, Writing – review and editing; Johnathon R Emlaw, Conceptualization, JRE acquired some of the initial recordings containing $\beta_{Anc}$ homopentamers., Writing – review and editing; Gregory D McCluskey, Conceptualization, Formal analysis, GDM reconstructed $\beta_{AncS}$ and acquired $\beta_{AncS}$ electrophysiological data., Investigation, Writing – review and editing; F Javier Pérez-Areales, FJPA acquired some of the initial recordings containing $\beta_{Anc}$ homopentamers., Investigation, Methodology, Writing – review and editing; Corrie JB daCosta, CJBdC interpreted the data and wrote the manuscript, and supervised the project., Conceptualization, Formal analysis, Funding acquisition, Investigation, Methodology, Project administration, Supervision, Writing – original draft, Writing – review and editing

## Author ORCIDs

Christian JG Tessier ⓘ http://orcid.org/0000-0002-6006-7755
Raymond M Sturgeon ⓘ http://orcid.org/0000-0002-0043-0241
Johnathon R Emlaw ⓘ http://orcid.org/0000-0001-7255-182X
Gregory D McCluskey ⓘ http://orcid.org/0000-0002-6594-4665
F Javier Pérez-Areales ⓘ http://orcid.org/0000-0001-9525-9346
Corrie JB daCosta ⓘ http://orcid.org/0000-0002-9546-5331

## Decision letter and Author response

Decision letter https://doi.org/10.7554/eLife.76504.sa1
Author response https://doi.org/10.7554/eLife.76504.sa2

---

# Additional files

## Supplementary files

• Supplementary file 1. Multiple sequence alignment in FASTA format used to identify residues that differ between $\beta_{Anc}$, $\beta_{AncS}$, and the human β-subunit, and that also align with residues shown to be important for homomeric pLGIC assembly (*Alves et al., 2011*; *Hannan and Smart, 2018*; *Sexton et al., 2021*; *Taylor et al., 1999*) and spontaneous activity (*Beckstead, 2002*; *Engel et al., 1996*; *Grosman and Auerbach, 2000*; *Miller et al., 2008*; *Mukhtasimova et al., 2009*; *Nayak et al., 2012*; *Ohno et al., 1995*; *Pan et al., 1997*; *Purohit and Auerbach, 2009*; *Torres and Weiss, 2002*).

• Transparent reporting form

## Data availability

All data generated and analysed during this study are included in the manuscript and as source data.

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

## Appendix 1

### Multiple sequence alignments

Provided is the multiple sequence alignment (*Supplementary file 1*) used to identify residues that differ between β$_{Anc}$, β$_{AncS}$, and the human β-subunit, and that also align with residues shown to be important for homomeric pLGIC assembly (*Alves et al., 2011*; *Hannan and Smart, 2018*; *Sexton et al., 2021*; *Taylor et al., 1999*) and spontaneous activity (*Beckstead, 2002*; *Engel et al., 1996*; *Grosman and Auerbach, 2000*; *Miller et al., 2008*; *Mukhtasimova et al., 2009*; *Nayak et al., 2012*; *Ohno et al., 1995*; *Pan et al., 1997*; *Purohit and Auerbach, 2009*; *Torres and Weiss, 2002*; *Appendix 1—figure 1*). Also shown is a local alignment of the second transmembrane region (M2) of the adult human muscle-type subunits (α-, β-, δ-, and ε-subunits), as well as β$_{Anc}$ and β$_{AncS}$ (*Appendix 1—figure 2*) highlighting differences in well-characterized determinants of single-channel conductance (*Imoto et al., 1991*; *Imoto et al., 1988*) and open-channel block (*Charnet et al., 1990*; *Leonard et al., 1988*; *Pascual and Karlin, 1998*).

**Appendix 1—figure 1.** Multiple sequence alignment of β$_{Anc}$ and β$_{AncS}$ with the human β-subunit (β$_{Hs}$). Positions aligning with those previously reported as being important for spontaneous openings (blue) and homomeric assembly (orange) in other pentameric ligand-gated ion channels (pLGICs) are highlighted. Sites implicated in both spontaneous opening and homomeric assembly are highlighted in yellow. In each case, if the residue implicated is conserved across all three subunits the highlight is faded. Corresponding references for each position are indicated (**a–n**), where a: *Torres and Weiss, 2002*; b: *Beckstead, 2002*; c: *Miller et al., 2008*; d: *Nayak et al., 2012*; e:*Purohit and Auerbach, 2009*; f: *Taylor et al., 1999*; g: *Sexton et al., 2021*; h: *Alves et al., 2011*; i: *Hannan and Smart, 2018*; j: *Pan et al., 1997*; k: *Mukhtasimova et al., 2009*; l: *Ohno et al., 1995*; m: *Grosman and Auerbach, 2000*; n: *Engel et al., 1996*.

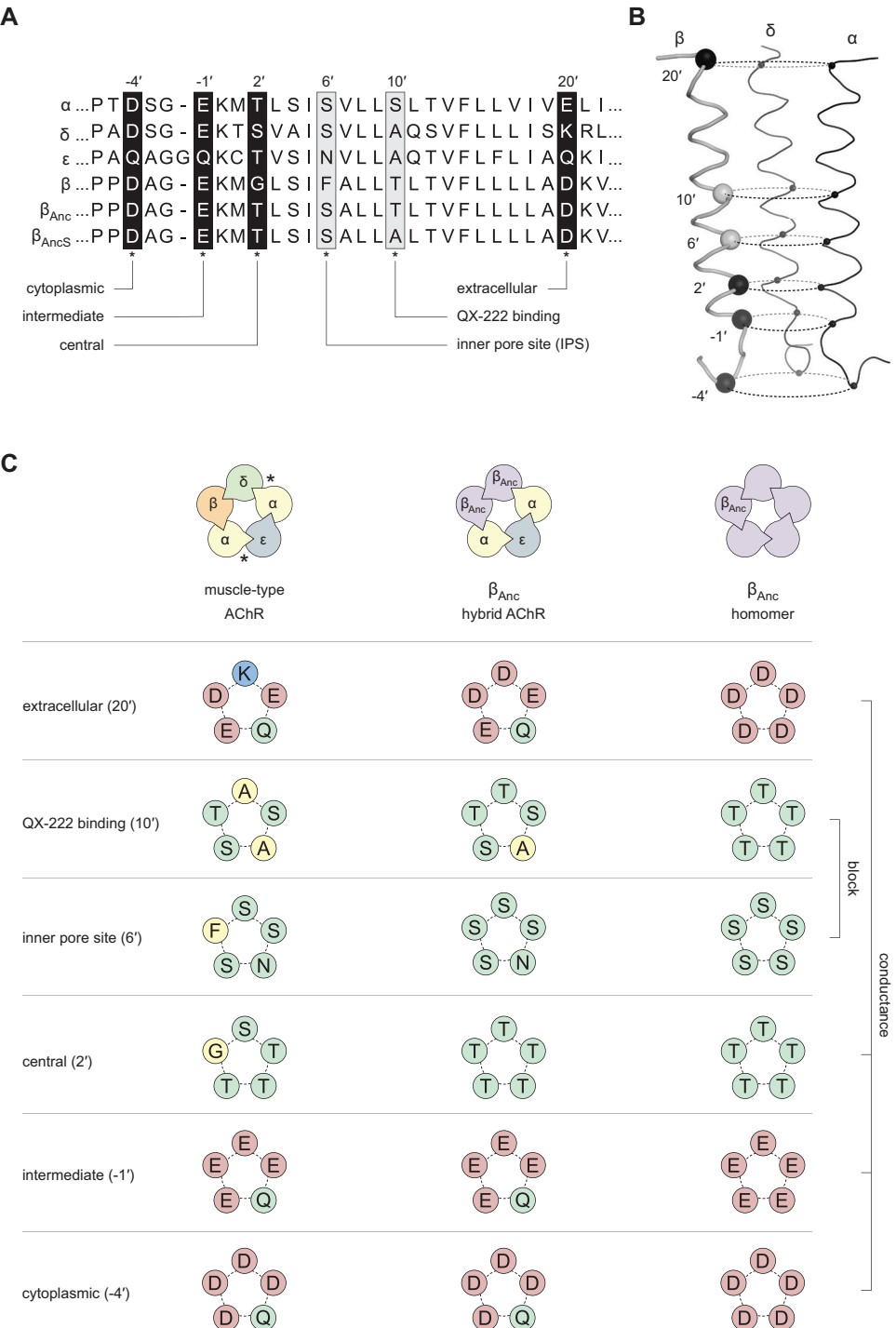

**Appendix 1—figure 2.** Primary and tertiary structure of the M2 segments from the adult human muscle-type AChR subunits, as well the two reconstructed ancestral β-subunits (β$_{Anc}$ and β$_{AncS}$). (**A**) Multiple sequence alignment of residues comprising the pore-lining M2 helix, highlighting positions discussed in the main text and known to influence single-channel conductance (black boxes; −4′, −1′, 2′, and 20′), or open-channel block (grey boxes; 6′ and 10′). (**B**) Tertiary structure of the AChR pore. M2 segments corresponding to the residues shown in panel "A" are mapped onto the Cα backbone of the *Torpedo* AChR (PDB: 6UWZ). The two subunits in the front (α and ε) have been hidden for illustrative purposes, and positions of highlighted residues in "A" are mapped onto the β-subunit with their Cα shown as black and grey spheres. (**C**) Analogous residues from each subunit form concentric rings, where the chemical properties of amino acids in each ring influence single-channel conductance or open-channel block. Residues contributed to each ring by individual subunits in the
*Appendix 1—figure 2 continued on next page*

*Appendix 1—figure 2 continued*

heteropentameric human adult muscle-type AChR (left) and $\beta_{Anc}$-containing hybrid AChR (middle), as well as in $\beta_{Anc}$ homompentamers (right) are shown. Acidic, basic, polar, and nonpolar residues are colored red, blue, green, and yellow, respectively.

