## [Editor Report]

This compelling study outlines how phylogenetic reconstruction can yield important insights into evolutionary aspects of assembly and function in acetylcholine receptors (AChRs). The authors elegantly demonstrate that ancestral AChR β subunits can form homomeric channels that share important functional hallmarks with their modern cousins, including unliganded gating. The work provides an intriguing framework to evaluate the evolution of the broader family of pentameric ligand-gated ion channels.

---

## [Decision Letter]

**Decision letter after peer review:**

Thank you for submitting your article "Ancestral acetylcholine receptor β-subunit forms homopentamers that prime before opening spontaneously" for consideration by *eLife*. Your article has been reviewed by 3 peer reviewers, and the evaluation has been overseen by a Reviewing Editor and Richard Aldrich as the Senior Editor. The following individuals involved in review of your submission have agreed to reveal their identity: Lucia Silvilotti (Reviewer #2); Wayland WL Cheng (Reviewer #3).

Essential revisions:

While the reviewers agree that no additional experiments are necessary, they would like the authors to carefully address all of the comments brought up in their respective reviews. Particular attention should be paid to the following aspects:

1) The work could be improved by providing additional mechanistic insight and discussion, specifically into the basis of homomeric assembly and unliganded gating in the ancestral channels. Please add a careful analysis and/or discussion regarding the sequence/structure of these ancestral proteins that may explain homomeric assembly and spontaneous opening. This could be aided by e.g. the literature on gain-of-function mutations and subunit assembly etc.

2) To make the manuscript accessible to a broad readership, please ensure that the main research question in clearly communicated throughout the manuscript, and include a clearer discussion and explanation of the potential evolutionary implications. Please see "Public review" by reviewer #1 and final comment by reviewer #2 in "Recommendations for the authors" for details.

*Reviewer #1 (Recommendations for the authors):*

The scientific questions asked by this study, and thus what we should take from this study, are not entirely clear, perhaps because the work emerged from a surprising finding in a closer examination of their earlier work. If the question is "did ancestral homo-pentamers evolve into α homo-pentamers and β homopentamers that then gave rise to obligate α/β heteropentamers?", the results would be interesting but require some analysis of amino acid sequences, both regarding receptor assembly and channel gating: this would be aided by a wealth of such analyses on acetylcholine receptors available in the literature. If the question is "did intermediate gating steps evolve before or after agonist binding?" then I don't think this has been adequately addressed, as β subunits, shown here to contain the intermediate steps but lack the requirement for agonist binding, evolved (1)long after agonist binding had emerged and (2) long after after acetylcholine receptors had diverged from other (homomeric) pLGICs that show similar intermediate steps.

Lines 201-209

It sounds reasonable/consistent with other work that increasing concentrations of acetylcholine would block the channel. But perhaps this interpretation deserves more detailed explanation, because I do not see how exactly the data in Figure 4 are consistent with the effects of channel block on dwell time histograms in Ref. 25. There is also a decrease in apparent current amplitude in the presence of QX-222, consistent with the blocker terminating the current before reaching its proper value (ref. 25) which I don't see for current in the presence of ACh. And while I understand that block by ACh might occur at lower concentrations than activation (or desensitization) by ACh, does block occur at concentrations as low as 10 μM? The analysis in Figure 4 (schemes) and table S2 seems to show that block is indeed similar for both ACh and QX-222, but I think the paragraph in question could be spelled out a little, given the focus of this paper on the activation mechanism.

*Reviewer #2 (Recommendations for the authors):*

The electrophysiology and its analysis is of high quality and supports well the conclusions. I would just like perhaps a bit more discussion of some points.

The discussion of why the conductance of the homomeric β receptors is higher than that of the heteromeric nAChR is quite short, it just says that it is to do with M2. Given that we know quite a bit of the conductance determinants of these channels, it ought to be possible to have more detail on this.

Equally, can you make sense of the similarity in channel block between homomers and heteromers on the basis of the pore lining M2 sequence?

Why do these channels open spontaneously? Is there any obvious gain-of-function "mutation"?

I accept that the evolutionary questions (last sentence of the public review) are very hard/impossible to answer, but a clearer discussion and explanation of these issues would be welcome.

*Reviewer #3 (Recommendations for the authors):*

The authors may consider adding the following to their discussion. In page 17 lines 329-331, they state that the results for β-Anc and β-AncS show that the functional hallmarks of flipped/primed states in the AchR do not arise from heteropentameric architecture or the presence of agonist. While this is a reasonable statement, the results from this study are not the only evidence to support these claims. There is evidence for primed states in other homopentameric ligand-gated ion channels, and in muscle-type acetylcholine receptor mutants that exhibit unliganded activity (reference 24 and other studies).

---

## [Author Response]

Essential revisions:While the reviewers agree that no additional experiments are necessary, they would like the authors to carefully address all of the comments brought up in their respective reviews. Particular attention should be paid to the following aspects:1) The work could be improved by providing additional mechanistic insight and discussion, specifically into the basis of homomeric assembly and unliganded gating in the ancestral channels. Please add a careful analysis and/or discussion regarding the sequence/structure of these ancestral proteins that may explain homomeric assembly and spontaneous opening. This could be aided by e.g. the literature on gain-of-function mutations and subunit assembly etc.

We agree. Our revised manuscript includes the requested analysis (see Appendix 1–figure 1 and Appendix 1–figure 2), as well as explicit discussion (lines 302-312) regarding “homomeric assembly and unliganded gating in the ancestral channels”.

2) To make the manuscript accessible to a broad readership, please ensure that the main research question in clearly communicated throughout the manuscript, and include a clearer discussion and explanation of the potential evolutionary implications.

The reviewers make two suggestions to increase accessibility to a broad readership:

“…ensure that the main research question in clearly communicated throughout the manuscript,”

We agree that this could have been clearer in our initial submission. We now explicitly state the motivation for this work in both the first paragraph of our Results and Discussion sections (lines 102-104 and 250-260).

“…include a clearer discussion and explanation of the potential evolutionary implications.”

At the request of the reviewers, we have added two paragraphs to the Discussion section explaining potential evolutionary implications (lines 360-395). In addition, where we mention these implications throughout the text, we have tried to make this explicitly clear (for example see the addition to lines 269-270).

Reviewer #1 (Recommendations for the authors):The scientific questions asked by this study, and thus what we should take from this study, are not entirely clear, perhaps because the work emerged from a surprising finding in a closer examination of their earlier work. If the question is "did ancestral homo-pentamers evolve into α homo-pentamers and β homno-pentamers that then gave rise to obligate α β hetero-pentamers?", the results would be interesting but require some analysis of amino acid sequences, both regarding receptor assembly and channel gating: this would be aided by a wealth of such analyses on acetylcholine receptors available in the literature. If the question is "did intermediate gating steps evolve before or after agonist binding?" then I don't think this has been adequately addressed, as β subunits, shown here to contain the intermediate steps but lack the requirement for agonist binding, evolved (1)long after agonist binding had emerged and (2) long after after acetylcholine receptors had diverged from other (homomeric) pLGICs that show similar intermediate steps.

We now make the motivation for the present work explicitly clear in the first paragraph of both the Results and Discussion sections of our revised manuscript. We also point out that the heterogeneity in our single-channel recordings was not present in earlier, published, recordings, and only became evident upon modification of our transfection protocol aimed at reducing the overall expression of β_Anc_-containing AChRs to facilitate kinetic analysis. Thus, this work does not stem from “a surprising finding in a *closer examination* of their (our) earlier work”, but instead from new single-channel data exhibiting heterogeneity that was not present in our previous work.

We agree that the present work does not uncover the complete and complex evolutionary history of α- and β-subunits, as well as the rise of α/β (and other) heteropentamers. We have now added explicit discussion of what we think the potential evolutionary implications of our findings are (lines 360-395).

We also agree that our findings do not uncover whether “intermediate gating steps evolve(d) before or after agonist binding”. Our findings demonstrate (as have others; see PMID: 19339970) that intermediate gating steps occur in the absence of agonist. In our case we also show, for the first time, that intermediate gating steps occur in a channel that is devoid (presumably) of agonist-binding sites (i.e. β_Anc_ homomers).

Lines 201-209It sounds reasonable/consistent with other work that increasing concentrations of acetylcholine would block the channel. But perhaps this interpretation deserves more detailed explanation, because I do not see how exactly the data in Figure 4 are consistent with the effects of channel block on dwell time histograms in Ref. 25. There is also a decrease in apparent current amplitude in the presence of QX-222, consistent with the blocker terminating the current before reaching its proper value (ref. 25) which I don't see for current in the presence of ACh. And while I understand that block by ACh might occur at lower concentrations than activation (or desensitization) by ACh, does block occur at concentrations as low as 10 μM? The analysis in Figure 4 (schemes) and table S2 seems to show that block is indeed similar for both ACh and QX-222, but I think the paragraph in question could be spelled out a little, given the focus of this paper on the activation mechanism.

We agree that as pointed out by the reviewer, the original Ref. 25 was inappropriate in the given context. In our revised document we correct this and provide several, more appropriate, references to support the statement that the observed “single-channel behaviour is a hallmark of open-channel block”.

The reviewer mentions that “there is also a decrease in apparent current amplitude in the presence of QX-222”. Motivated by this comment we have examined this apparent effect in detail, and while we agree that the individual burst originally shown in Figure 4B (at 60 μM QX-222) has a visibly lower amplitude, the effect is not robust across 30 bursts (from three recordings), as well as additional recordings made at –70 mV (not shown). Given this, and Neher’s 1983 work on QX-222 block of frog endplate AChRs (PMID: 6310093), where he showed that the “amplitude of the individual current pulses was not changed by the presence of the drug up to a concentration of 250 μM” we cannot make the case that this is a legitimate phenomenon for either adult human wild-type AChRs or β_Anc_ homopentamers. For these reasons, we have refrained from discussing this potential effect of the blocker. In addition, to avoid leading the reader to a similar conclusion as the reviewer, we have replaced the original burst in question in Figure 4B with a burst that has an amplitude more representative of the mean amplitude for this condition (from 30 bursts from a total of three recordings).

Finally, as requested by the reviewer we have expanded our explanation of how block is similar for both ACh and QX-222 in the wild-type AChR and β_Anc_ homomers in both the Results (lines 170-175) and Discussion sections (lines 340-359).

Reviewer #2 (Recommendations for the authors):The electrophysiology and its analysis is of high quality and supports well the conclusions. I would just like perhaps a bit more discussion of some points.The discussion of why the conductance of the homomeric β receptors is higher than that of the heteromeric nAChR is quite short, it just says that it is to do with M2. Given that we know quite a bit of the conductance determinants of these channels, it ought to be possible to have more detail on this.Equally, can you make sense of the similarity in channel block between homomers and heteromers on the basis of the pore lining M2 sequence?

We agree. Our revised manuscript now contains extensive discussion, and an additional Appendix figure (Appendix 1–figure 2), detailing why the conductance of β_Anc_ homomers is higher than that of the heteromeric AChR (lines 313-413), as well as why block is similar between the two types of channels (lines 340-359).

Why do these channels open spontaneously? Is there any obvious gain-of-function "mutation"?

We agree that this is an important question. At present it is unclear why these channels open spontaneously. We now explicitly mention in the main text that pinpointing the residue (or set of residues) necessary is not trivial (lines 376-386), and identifying these residues is beyond the scope of the present work. Nevertheless, to facilitate preliminary hypotheses, our revised manuscript now contains a multiple sequence alignment highlighting positions that differ between β_Anc_ (or β_AncS_) and the human β-subunit, and that have also been shown to lead to spontaneous activity in other pLGICs (see Appendix 1–figure 1).

I accept that the evolutionary questions (last sentence of the public review) are very hard/impossible to answer, but a clearer discussion and explanation of these issues would be welcome.

At the request of the reviewer, we have added two paragraphs to the Discussion section explaining potential evolutionary implications (lines 360-395).

Reviewer #3 (Recommendations for the authors):The authors may consider adding the following to their discussion. In page 17 lines 329-331, they state that the results for β-Anc and β-AncS show that the functional hallmarks of flipped/primed states in the AchR do not arise from heteropentameric architecture or the presence of agonist. While this is a reasonable statement, the results from this study are not the only evidence to support these claims. There is evidence for primed states in other homopentameric ligand-gated ion channels, and in muscle-type acetylcholine receptor mutants that exhibit unliganded activity (reference 24 and other studies).

We agree that there is accumulating evidence for intermediate shut states preceding channel opening in a host of hetero- and homomeric pLGICs (as well as other families of LGICs), but it is unclear whether the majority of these states represent the same “flipped” and “primed” states originally inferred from single-channel data (see for example PMID: 33567265). Given that the intermediate states we detect also derive from single-channel data, we think it is appropriate to restrict our discussion to these directly analogous states.

With regards to additional evidence that “primed states” occur “in muscle-type acetylcholine receptor mutants that exhibit unliganded activity”, we agree and have referenced this (with the specific reference the reviewer suggests) in the final paragraph of our Results section (line 242).